# Chalcone T4 Inhibits RANKL-Induced Osteoclastogenesis and Stimulates Osteogenesis In Vitro

**DOI:** 10.3390/ijms24087624

**Published:** 2023-04-21

**Authors:** Iolanda Augusta Fernandes de Matos, Natalie Aparecida Rodrigues Fernandes, Giovani Cirelli, Mariely Araújo de Godoi, Letícia Ribeiro de Assis, Luis Octávio Regasini, Carlos Rossa Junior, Morgana Rodrigues Guimarães-Stabili

**Affiliations:** 1Department of Diagnosis and Surgery, School of Dentistry of Araraquara, São Paulo State University (UNESP), Araraquara 14801-385, SP, Brazil; iolanda.matos@unesp.br (I.A.F.d.M.); natalie.fernandes@unesp.br (N.A.R.F.);; 2Department of Chemistry and Environmental Sciences, Institute of Biosciences, Humanities and Exact Sciences, São Paulo State University (UNESP), São José do Rio Preto 15054-000, SP, Brazil

**Keywords:** chalcone, osteoclastogenesis, osteogenesis, intracellular signaling

## Abstract

Chalcones are phenolic compounds produced during the biosynthesis of flavonoids that have numerous biological activities, including anti-inflammatory, antioxidant and anticancer. In this in vitro study, we investigate a newly synthesized chalcone (Chalcone T4) in the context of bone turnover, specifically on the modulation of osteoclast differentiation and activity and osteoblast differentiation. Murine macrophages (RAW 264.7) and pre-osteoblasts (MC3T3-E1) were used as models of osteoclasts and osteoblasts, respectively. Differentiation and activity osteoclasts were induced by RANKL in the presence and absence of non-cytotoxic concentrations of Chalcone T4, added in different periods during osteoclastogenesis. Osteoclast differentiation and activity were assessed by actin ring formation and resorption pit assay, respectively. Expression of osteoclast-specific markers (*Nfatc1*, *Oscar*, *Acp5*, *Mmp-9* and *Ctsk*) was determined by RT-qPCR, and the activation status of relevant intracellular signaling pathways (MAPK, AKT and NF-kB) by Western blot. Osteoblast differentiation and activity was induced by osteogenic culture medium in the presence and absence of the same concentrations of Chalcone T4. Outcomes assessed were the formation of mineralization nodules via alizarin red staining and the expression of osteoblast-related genes (*Alp* e *Runx2*) by RT-qPCR. Chalcone T4 reduced RANKL-induced osteoclast differentiation and activity, suppressed *Oscar*, *Acp5* and *Mmp-9* expression, and decreased ERK and AKT activation in a dose-dependent manner. *Nfact1* expression and NF-kB phosphorylation were not modulated by the compound. Mineralized matrix formation and the expression of *Alp* and *Runx2* by MC3T3-E1 cells were markedly stimulated by Chalcone T4. Collectively, these results demonstrate that Chalcone T4 inhibits in osteoclast differentiation and activity and stimulates osteogenesis, which indicates a promising therapeutic potential in osteolytic diseases.

## 1. Introduction

Bone loss is among the major problems associated with aging, rheumatoid arthritis, osteoporosis, cancer, periodontitis, and other chronic inflammatory diseases [1]. This has driven the investigation of molecular mechanisms controlling bone turnover. Activation of osteoclasts, multinucleated cells derived from hematopoietic cells, is regulated by several molecular signals triggered by the activation of receptor activator of nuclear factor kappa beta (RANK) by its cognate molecule, receptor activator of nuclear factor kappa beta ligand (RANKL). RANK/RANKL interaction leads to the activation of tumor necrosis factor receptor associated factor 6 (TRAF6) intracellularly and ultimately induces activation of various other signaling intermediates and transcription factors, including c-Fos and nuclear factor of activated T cells (NFATc1), mitogen-activated protein kinases (MAPKs), nuclear factor kappa-light-chain-enhancer of activated B cells (NF-kB), c-Src, phosphatidylinositol-3 kinase (PI3K) and protein kinase B (Akt) [2]. Activated NFATc1 can increase the expression of specific markers of osteoclastogenesis, such as tartrate-resistant acid phosphatase (TRAP), cathepsin K (CTSK), and matrix metalloproteinase (MMP-9), which act at various stages of osteoclast development and activation and mediate degradation of inorganic and organic components of bone [3].

Multiple pharmacologic approaches have been developed for osteolytic diseases and conditions, many of these are currently used clinically, including antiresorptive drugs such as bisphosphonates [4], hormone replacement therapies [5], low-dose doxycycline [6], macromolecular inhibitors (such as monoclonal antibodies [3]), however, the long-term use of these drugs has been associated with deleterious secondary effects, which stresses the need for the development of alternative therapies.

The use of natural compounds or synthetic compounds based on their natural prototypes has aroused great scientific interest due to their biological activities and low toxicity. Among these compounds, the chalcones, a group of phenolic substances that are precursors of the synthesis of flavonoids, have been highlighted for their ability to inhibit osteoclastogenesis in vitro and bone resorption in vivo [7,8].

Chalcones can modulate several targets of the inflammatory process, including transcription factors, intracellular signaling pathways and inflammatory mediators [9,10], which are essential in the process of osteoclast activation and differentiation. This biological property makes Chalcones an interesting therapeutic option for both inflammatory diseases (e.g., inflammatory bowel diseases, rheumatoid arthritis, sepsis) [11,12,13,14,15] and osteolytic diseases (bone cancer, osteoporosis and periodontitis) [16,17,18].

Corroborating evidence indicating the anti-osteolytic properties of chalcones, our research group recently demonstrated the anti-inflammatory and anti-resorptive effects of a new synthetic chalcone, Chalcone T4, on inflammation and resorption bone in an experimental model of periodontitis in rats [7]. The systemic administration of the compound, tested for the first time, reduced the cellular infiltrate in the gingival tissues and inhibited the ligature-induced alveolar bone resorption, without causing unwanted secondary effects [7]. These promising in vivo results demonstrating the antiresorptive potential of Chalcone T4 spurred us on to investigate the possible biological mechanisms involved, as the comprehension of its biological activities on bone turnover can indicate its potential in other osteolytic diseases.

In the present study, we studied the influence of Chalcone T4 on bone coupling and turnover by investigating in vitro its effects on osteoclast differentiation and activity and the osteogenic potential. We also assessed the regulation of candidate molecular targets to gain insight into biological mechanisms mediating the effects of Chalcone T4.

## 2. Results

### 2.1. Chalcone T4 Inhibits Osteoclast Differentiation and Activity in a Dose-Dependent Manner

To evaluate the possibilities of a dose–response effect and of distinct effects of Chalcone T4 on osteoclast differentiation, maturation and activity, the compound was added to the culture medium at different periods in relation to stimulation with RANKL and also in different concentrations (1–10 µM). Stimulation with RANKL induced differentiation of macrophages into osteoclasts, which was inhibited by Chalcone T4 in a concentration-dependent manner, at all periods analyzed (Figure 1). Inhibition of RANKL-induced osteoclastogenesis by Chalcone T4 was more effective when the compound was added to the culture in the initial periods of the differentiation process (30 min before and 48 h after stimulation with RANKL). Notably, the compound markedly inhibited osteoclastogenesis even when added at the lowest concentration (1 μM) (Figure 1(AI,AII,BI,BII), and completely suppressed differentiation when used at higher concentrations (5 and 10 Μm) (Figure 1(AI,BI). Interestingly, even when added at later stages of osteoclastogenesis (96 h after RANKL stimulation), Chalcone T4 was able to block differentiation at concentrations greater than 1 μM (*p* < 0.05) (Figure 1(AIII,BIII)).

In parallel with the effects of Chalcone T4 on osteoclastogenesis, we also determined its effects on osteoclast activity. In the pit assay, cells are plated in calcium phosphate-coated culture wells, and osteoclast resorptive activity is proportional to the areas cleared of the coating (i.e., areas of exposed cell culture plastic). In this assay, stimulation with RANKL induced a significant resorptive activity (control vs. RANKL, *p* < 0.05), (Figure 2), which was significantly reduced by Chalcone T4 added previously to RANKL stimulation at concentrations of 2.5 μM or greater (Figure 2(AI,BI)). When added after RANKL stimulation, only the highest concentration (10 μM) of Chalcone T4 inhibited resorptive activity of osteoclasts significantly (Figure 2(AII,AIII,BII,BIII)).

### 2.2. Chalcone T4 Reduces Expression of Osteoclastogenesis Markers, and Inhibits RANKL-Induced Activation of ERK and AKT

To obtain insight into the molecular mechanisms mediating Chalcone T4 effects on osteoclastogenesis, we assessed gene expression of selected candidate osteoclast markers (*Nfatc1*, *Oscar*, *Acp5*, *Mmp-9* and *Cathepsin-k*), and the activation of relevant signaling pathways (ERK, p38, AKT and NF-kB). Stimulation with RANKL increased the expression of all selected candidate genes and Chalcone T4 concentration-dependently reduced the expression of *Mmp-9*, *Cathepsin K* and *Oscar* (Figure 3) at both 3 and 5 days after RANKL stimulation. Chalcone T4 reduced *Acp5* transcript levels only at 5 days after RANKL stimulation, whereas expression of *Nfatc1* was not significantly affected in either period (Figure 3).

All signaling pathways assessed were activated by stimulation with RANKL (Figure 4). The highest concentrations of chalcone T4 significantly inhibited activation of ERK (10 μM) and AKT (5 μM and 10 μM). Interestingly, inhibition of these signaling pathways was observed at different dynamics after RANKL stimulation: 10 min for ERK and 40 min for AKT (Figure 4A,B).

### 2.3. Chalcone T4 Stimulates Formation of Mineralized Matrix and Expression of Osteogenic Markers

Coupled bone turnover involves a careful balance between bone formation by osteoblasts and bone resorption by osteoclasts. Thus, we investigated if the bone sparing effect observed in vivo in our previous study [7] was also associated with increased bone formation by assessing the effects of Chalcone T4 on the formation of mineralized nodules and expression of osteoblastic differentiation markers by pre-osteoblastic MC3T3-E1 cells. When used in concentrations of 2.5 µM or more, Chalcone T4 increased mineralization activity on the 21st day of differentiation (Figure 5A) and also significantly increased expression of osteogenic markers *Alp* and *Runx2* (Figure 5B).

## 3. Discussion

Our group demonstrated in vivo the bone sparing effect of Chalcone T4 reducing inflammatory bone resorption in an experimental periodontitis model in rats [7]; however, the cellular and molecular mechanisms involved were not explored. Considering that this knowledge may indicate the potential of the compound in the therapy of other osteolytic diseases and conditions, we investigated the effects of Chalcone T4 on RANKL-induced differentiation and resorptive activity of osteoclasts in vitro, assessing its effects on the expression of selected candidate osteoclast marker genes and also on the activation of key intracellular signaling pathways. Since coupled bone turnover is the balance between bone formation and resorption, we extended our investigation to assess the influence of Chalcone T4 on osteoblast differentiation and activity. Chalcone T4 inhibited the differentiation of precursor cells into osteoclasts and markedly reduced the resorptive activity of these cells, even when added at late stages of RANKL-induced osteoclastogenesis. These results were associated with downregulation of expression of the selected osteoclast marker genes and inhibition of ERK and AKT signaling induced by RANKL stimulation. Notably, Chalcone T4 also enhanced osteoblast differentiation and activity in vitro by increasing both the expression of *Runx2* and *Alp* and the formation of mineralized nodules.

MAPKs, NF-κB and AKT signaling pathways are critical for osteoclast differentiation and function. These pathways are activated by a variety of cytokines, including RANKL [19,20,21,22]. The rapid and transient activation of these signaling pathways will ultimately regulate the expression of genes necessary for osteoclast differentiation [23]. To gain insight into the biological mechanism involved in the effects of Chalcone T4 on osteoclastogenesis, we determined its influence on RANKL-induced activation of these signaling pathways. Chalcone T4 reduced RANKL-induced ERK and AKT activation with different temporal dynamics.

ERK activation is considered crucial for osteoclast survival, and that this transcription factor can positively regulate osteoclast differentiation, migration, and resorption activity in vitro, in addition to interfering with bone metabolism in vivo [19,20,21]. Similarly, results have also shown that the AKT signaling pathway plays an important role in osteoclastogenesis, in addition to directly stimulating osteoblast differentiation and function [22]. Osteoclasts derived from bone marrow cells of animals genetically deficient for AKT showed an inhibition in the cellular differentiation and activity, a phenomenon correlated with the inhibition of the production of RANKL and OPG [23]. This information supports a role for ERK and AKT in osteoclastogenesis and our results suggest that the modulatory effects of Chalcone T4 on osteoclast differentiation and activity may be related, at least in part, to RANKL-induced inactivation of these pathways. The reduction in gene expression of markers of osteoclast differentiation and function (*Mmp-9*, *Acp5* and *Oscar*) by Chalcone T4 is also in line with the downregulation of ERK and AKT, since activation of these signaling pathways can increase transcription of these genes.

Several studies have also shown that activation of the NF-ĸB pathway can favor osteoclast survival and bone resorption, in addition to activating transcription factors that are essential for osteoclast differentiation such as NFATc1 and c-Fos [24]. In vitro and in vivo studies demonstrating inhibition of osteoclastogenesis by chalcones have shown that inhibition of NF-ĸB by these compounds has a critical role [24]. However, despite the importance of NF-kB for osteoclast differentiation, Chalcone T4, which effectively reduced osteoclastogenesis and activity, did not inhibit RANKL-induced NF-kB activation in the experimental periods assessed.

Interestingly, the reduction in osteoclast differentiation and activity by Chalcone T4 was also not associated with a significant reduction in *Nfatc1* gene expression by compound. It is important to consider the limitations of the experimental approach (mRNA only), and it is possible that Chalcone T4 had post-transcriptional effects, including changes in phosphorylation/subcellular localization of NFATc1. It is also possible that NFATc1-independent mediators and/or mechanisms may be involved in this anti-osteolytic effect of Chalcone T4. 

The anti-osteoclastogenic effect of Chalcone T4 is congruent with other in vitro studies demonstrating anti-osteolytic effects of chalconic compounds. In these studies, the anti-resorptive effect has been associated with their ability to modulate transcription factors [25,26], inflammatory mediators [25], and intracellular signaling pathways [27] involved in cell differentiation and activity of osteoclasts. Reduction in mRNA levels and protein of *Cathepsin k*, *c-fos*, *Traf6* and *Trap* were observed in macrophages treated with natural chalcones and corresponded to reduced differentiation and activity of osteoclasts in vitro [26]. Modulation of MAPKs by chalcones has also been observed and associated with their effects on osteoclastogenesis [28]. In most of these studies, however, these biological effects of chalconic compounds are observed in higher concentrations than those used in this present study with the novel Chalcone T4, and/or only when Chalcone treatment was implemented at the onset of osteoclastogenesis. In fact, the potency and duration of the biological effects of synthetic chalcones have been shown to be many times superior than those of natural chalcones due to structural modifications and changes in their physical-chemical properties [29]. 

Based on the inhibitory effects of Chalcone T4 on osteoclasts and considering that bone turnover is a coupled process of resorption and synthesis/formation, this study also evaluated its effect on osteoblastic differentiation and activity. Formation of mineralized nodules by osteoblasts in vitro is a prototypical method for evaluating osteoblast activity, and the expression of osteoblast-related genes *Alp* and *Runx2* depicts osteoblastic differentiation [30]. Chalcone T4 increased both the formation of mineralized nodules and the expression of *Alp* and *Runx2*, indicating its potential biological effect in osteogenesis. A few studies have reported on the osteogenic effect of natural chalcones in vitro, demonstrating their potential to increase the proliferation and differentiation of pre-osteoblastic cells in both MC3T3-E1 cells [28] and bone marrow-derived mesenchymal stem cells (BMSCs) [31]. However, these studies also used with chalcone concentrations much higher than those of Chalcone T4 used in this study, suggesting a greater potency of this novel compound. Biological mechanisms suggested for the osteogenic effects of chalcones include the increase in gene expression of osteogenic markers and the activation of intracellular signaling pathways important in cell proliferation and differentiation in osteoblasts (e.g., bone morphogenetic protein (BMP) pathways, Wnt/Beta-catenin, and MAPKs) [31].

Collectively, our results demonstrate that low concentrations of Chalcone T4 significantly suppressed RANKL-induced osteoclastic differentiation and activity in macrophages and stimulated mineralized matrix formation and mRNA levels of osteogenic markers in osteoblasts in vitro. Considering the low cost and ease of the synthesis, the absence of toxicity and its effectiveness in reduced concentrations, Chalcone T4 may be a promising therapeutic alternative in several osteolytic diseases. Additional studies exploring the role of Chalcone T4 on other transcription factors and mediators involved in osteoclast differentiation and activity should be conducted, as well as studies evaluating the effects of compound on osteoblast activity and differentiation, and on bone formation in different vivo models of osteolytic diseases and conditions.

## 4. Materials and Methods

### 4.1. Chalcone T4

Chalcone was synthesized by Dr. Luis Octavio Regasini, at the Laboratory of Chemistry and Environmental Sciences, at São Paulo State University (UNESP). All reagents and solvents were purchased from Merck (Rahway, NJ, USA). The synthesis of Chalcone T4 was performed according to the protocol described by Camargo et al. [32], with some modifications. The synthesis of the substance was carried out by means of the Claisen-Schmidt condensation reaction, with satisfactory yields (50–70%). For the purification of the substances, recrystallization and chromatography techniques were used, including normal phase column chromatography (silica gel), reversed phase column chromatography (octadecylsilane) and gel permeation chromatography (LH-20). The structure of the compound was identified by mass spectrometry and hydrogen and thirteen carbon nuclear magnetic resonance techniques (1H NMR and 13C NMR) (Appendix A). The purity of the substance was determined by high performance liquid chromatography analysis with photodiode array detection (HPLC-DAD), showing values equal to or greater than 95.0%.

### 4.2. Effect of Chalcone T4 on Osteoclast Differentiation

RAW 264.7, murine osteoclast precursor cells, were seeded (0.5 × 10^3^ cells/well) in a 96-well plate in α-MEM (Gibco, ThermoFisher Scientific, Waltham, MA, USA) supplemented (10% fetal bovine serum (FBS)) (Invitrogen, Carlsbad, CA, USA) + 1% penicillin/streptomycin (P/S) (Invitrogen, Carlsbad, CA, USA), in the presence of RANKL (#315-11, Peprotech, Cranbury, NJ, USA) (100 ng/mL) for differentiation of macrophages into osteoclasts. Different non-cytotoxic concentrations of Chalcone (1, 2.5, 5 and 10 μM) or vehicle (DMSO) (Sigma-Aldrich, San Luis, MO, USA) (Appendix A), were added to the culture medium at different times in relation to the differentiation stimulus (RANKL): 30 min before, 48 h and 96 h after. Culture medium containing the compound and RANKL were replaced at 48 h intervals, removing 100 μL and adding the same volume of fresh medium with RANKL and/or Chalcone T4. At the end of the experimental period (6th day), the cells were fixed and permeabilized by adding 50 µL of Cytofix/Cytoperm solution (BD Cytofix/Cytoperm, BD Biosciences, San Jose, CA, USA) per well. Cells were then stained with 5 μg/mL fluorophore-conjugated phalloidin (AlexaFluor 488 Phalloidin, Molecular Probes, ThermoFisher Scientific, Waltham MA, USA) for 40 min at room temperature, and subsequently with 2.5 μg/mL 4′,6-diamidine-2-phenylindole (DAPI) (Invitrogen, Carlsbad, CA, USA) 5 min at room temperature and visualized under an inverted fluorescence microscope (Evos fl, AMG Micro) in a 4× objective. The number of osteoclasts (large cells displaying actin ring formation and presenting 3 or more nuclei) was quantified over the entire wells, by a blind examiner.

### 4.3. Effect of Chalcone T4 on Osteoclast Activity

To investigate the impact of Chalcone T4 on osteoclast activity, RAW 264.7 were seeded in 96-well plates coated with synthetic calcium phosphate (Osteologic Corning Plate, Corning), and stimulation with RANKL and treatment with different concentrations of Chalcone T4 were performed as described above, for a period of 6 days, following the manufacturer’s instructions. On the sixth day, the supernatant was discarded, and 1% sodium hypochlorite was added for 5 min. The wells were washed with Milli-Q water, and the plates were left on the bench at room temperature for 24 h for complete evaporation of the water, and then photographed under an inverted microscope with an attached camera (Evos fl, AMG Micro, Seattle, WA, USA), in the 10× objective. Osteoclast activity was assessed by measuring the area of exposed cell culture plastic (i.e., the area in which the calcium phosphate coating was resorbed) in each well. Bone resorption areas were quantified along the entire length of the well using Image J software (http://rsbweb.nih.gov/ij/). The measurements were performed by a blind examiner to the experimental groups. 

### 4.4. Evaluation of Cytokine Gene Expression (RT-qPCR)

RAW 264.7 cells were seeded (3 × 10^5^ cells/well) in a 6-well plate and cultured in complete α-MEM in the presence of RANKL (100 ng/mL). To investigate the effect of the compound on the expression of markers related to osteoclastogenesis, different concentrations of Chalcone were added to the culture medium, 30 min prior to stimulation with RANKL, and kept for 3 and 5 days. The culture medium containing the compound and/or RANKL was replaced at 48h intervals. At the end of the third and fifth day, total RNA was isolated from cells with the Purification kit (Cellco Biotec), according to the manufacturer’s instructions. The quantification and purity of the RNA samples were performed in a nanovolume spectrophotometer (Nanoview Plus, GE Healthcare, Buckinghamshire, UK), and 500 ng were reverse transcribed into cDNA (High-capacity cDNA synthesis kit, Applied Biosystem). The detection of gene expression in the sample was performed using the TaqMan Fast Universal PCR Master Mix system (Applied Biosystems^TM^; Thermofisher Scientific, Waltham, MA, USA). PCR was performed to detect the target genes of interest (*Nfatc1*, osteoclast-associated immunoglobulin-like receptor (*Oscar*), acid phosphatase 5, tartrate Resistant (*Acp5)*, *Mmp-9* and *Ctsk*—Appendix A) and the expression of glyceraldehyde-3-phosphate dehydrogenase (*Gapdh*) was used to normalize the results, as a constitutive gene not affected by the experimental conditions. To compare the expression levels among different samples, the relative expression level of the genes was calculated using the comparative Δ(ΔCT) method using the thermocycler’s software [33]. The 3- and 5-day periods were selected from the preliminary experiments evaluating the periods of highest expression of target genes after RANKL stimulation.

### 4.5. Role of Chalcone in the Activation of Intracellular Signaling Pathways (Western Blot)

In order to investigate the effects of Chalcone T4 on the phosphorylation of intracellular signaling pathways NF-ĸB, p38 and ERK (MAPK), and AkT, macrophages were plated at a concentration of 3 × 10^5^ cells/well with α-MEM supplemented, and after 24 h de-induced with culture medium containing 0.3% FBS for 6 h, and then stimulated with RANKL for 10 and 40 min, with and without pretreatment with Chalcone T4. At the end of the experimental period, the medium was aspirated, the plates washed with PBS and RIPA Buffer (Sigma-Aldrich, San Luis, MO, USA) supplemented with phosphatase and protease inhibitor (Santa Cruz Biotechnology, Santa Cruz, CA, USA) added to each plate. Total proteins were quantified by the Bradford method (Bio-Rad Laboratories, Hercules, CA, USA). To perform the Western Blot, 20 μg of total protein per sample were used. The membranes were incubated with the primary antibodies (p-p38 #9211 Cell Signaling, p38 #9212 Cell Signaling, p-Akt #9275 Cell Signaling, Akt #9272 Cell Signaling, p-NF-kB #3033 Cell Signaling, NF-kB #sc8008 Santa Cruz, p-ERK #4376 Cell Signaling and ERK #9102 Cell Signaling) overnight. The results were obtained by densitometric analysis of the bands and normalized through the ratio between the phosphorylated and non-phosphorylated form of the target protein.

### 4.6. Effect of Chalcone T4 on Osteogenic Differentiation (Alizarin Red and RT-qPCR)

MC3T3-E1, pre-osteoblastic cells from mice (Sigma-Aldrich, San Luis, MO, USA) were plated in a 96-well plate, with 3 × 10³ cells per well, and cultured with α-MEM supplemented (10% FBS + 1% P/S), with different non-cytotoxic concentrations of Chalcone T4 (determined in preliminary experiments (Appendix A), for 7, 14 and 21 days. After the periods, the medium was aspirated, and the wells washed with PBS (Invitrogen, Carlsbad, CA, USA). The cells were fixed with 4% paraformaldehyde for 60 min at 4 °C, washed with distilled water, and stained with 2% alizarin red solution (Sigma-Aldrich, San Luis, MO, USA) for 15 min. After removing the dye, 10% cetylpyridinium chloride (Sigma-Aldrich, San Luis, MO, USA) was added for 10 min to solubilize the calcium nodules stained with alizarin red, and the solution transferred to a new 96-well plate for reading in a spectrophotometer at 550 nm. At the end of each experimental period, after staining with alizarin red, and prior to the addition of cetylpyridinium chloride, the images of each well were registered on a digital inverted microscope (brightfield), at a 40× magnification.

To evaluate the effect of Chalcone T4 on the expression of osteogenic markers, the cells were seeded (3 × 10^5^ cells/well) in a 6-well plate and maintained in osteogenic medium, with different concentrations of compound for 7 days. The culture medium containing the Chalcone T4 was replaced at 48h intervals. At the end of the seventh day, total RNA was isolated from cells and used for cDNA synthesis and expression analysis of target genes (*Alp* and *Runx2*-Appendix A) by RT-qPCR, according to the protocol described above.

### 4.7. Statistical Analysis

Data obtained from each assessment were analyzed using the statistical software GraphPad Prism 6 (GraphPad Software Inc., San Diego, CA, USA). The Kolmogorov–Smirnov test was applied to analyze the data distribution. One-way ANOVA and post hoc Tukey tests were performed for multiple comparison among the groups. All tests were applied with a 95% confidence level (*p* < 0.05). All experiments were performed at least three times in duplicate independently.

## Figures and Tables

**Figure 1 ijms-24-07624-f001:**
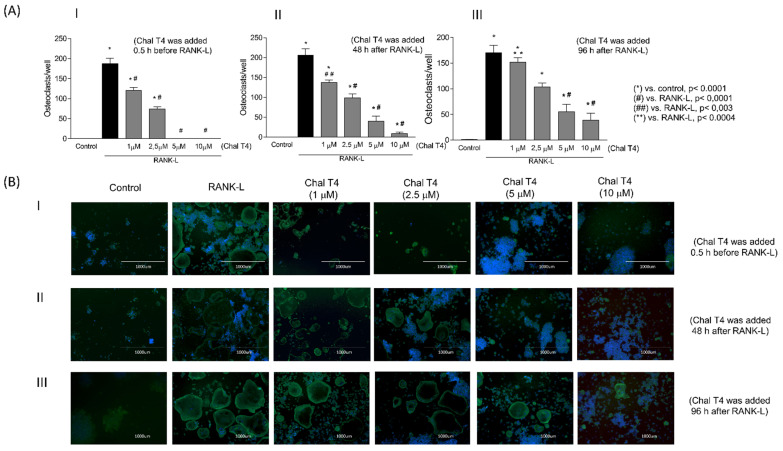
Chalcone T4 reduces osteoclast differentiation in a dose-dependent manner, even when added up to 96 h after RANKL stimulation. Cells were seeded and stimulated with RANKL (100 ng/mL) and/or treated with different concentrations of Chalcone T4. In (**A**) The bar graph represents the number of osteoclasts per well after stimulation with RANKL, and treatment with Chalcone at different times ((**I**): 30 min before stimulation with RANKL, (**II**): 48 h and (**III**): 96 h after stimulation). In (**B**) Images obtained in fluorescence microscopy (4× magnification), representative of the differentiation of osteoclasts induced by RANKL, after treatment or not of cells with Chalcone T4. The bars indicate the mean values and the vertical lines the standard error of the mean (SEM) of three independent experiments, performed in duplicate. *p* values are depicted in the figure.

**Figure 2 ijms-24-07624-f002:**
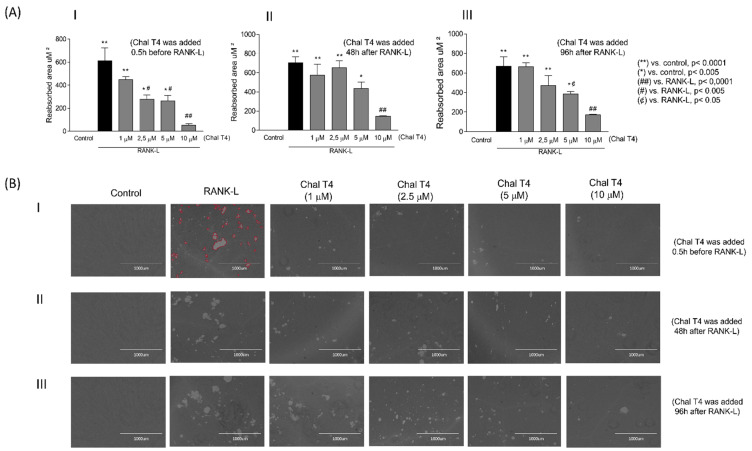
Chalcone T4 inhibits osteoclast activity in a dose-dependent manner, in all analyzed periods. Cells were stimulated with RANKL and treated with chalcone as described above. The resorptive activity of osteoclasts was evaluated through the area of the resorption gaps (underlined in red) in the plates coated with synthetic calcium phosphate. (**A**) (**I**–**III**) quantitative result of the area reabsorbed by osteoclasts, in the different periods of chalcone addition. (**B**) (**I**–**III**) representative images of the area of resorption after stimulation of cells with RANKL and treatment with the compound (4× magnification). The bars indicate the mean values and the vertical lines the standard error of the mean (SEM) of three different experiments evaluated in triplicate. Statistically significant differences between groups, as well as *p* values are indicated in the figure.

**Figure 3 ijms-24-07624-f003:**
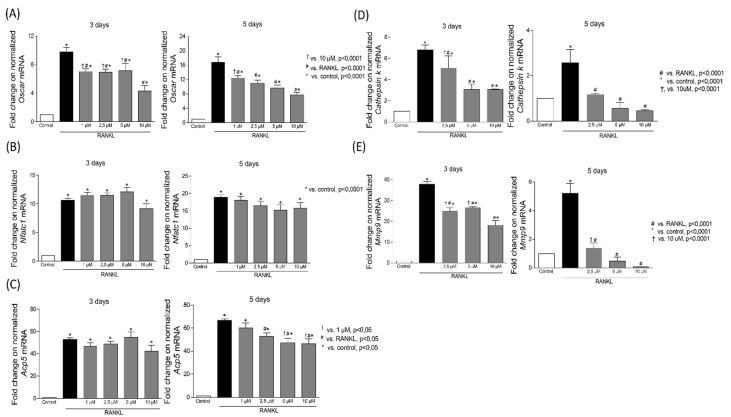
Chalcone T4 inhibits gene expression of bone markers in RANKL-stimulated RAW 264.7 cells. Effect of chalcone T4 on gene expression of *Oscar* (**A**), *Nfatc1* (**B**), *Acp5* (**C**), *Cathepsin k* (**D**) and *Mmp-9* (**E**). Chalcone significantly inhibited the gene expression of *Mmp-9*, *Cathepsin k* and *Oscar* on the third and fifth day after stimulation with RANKL (*p* < 0.0001), and only in the later period (fifth day) the expression of *Acp5*. *Nfatc1* expression was not altered by Chalcone treatment. Data is presented as fold change to unstimulated (control) cells. The bars indicate the mean results and the vertical lines the standard error (SEM) of three independent experiments performed in duplicate. Statistically significant differences between groups, as well as *p* values are indicated in the figure.

**Figure 4 ijms-24-07624-f004:**
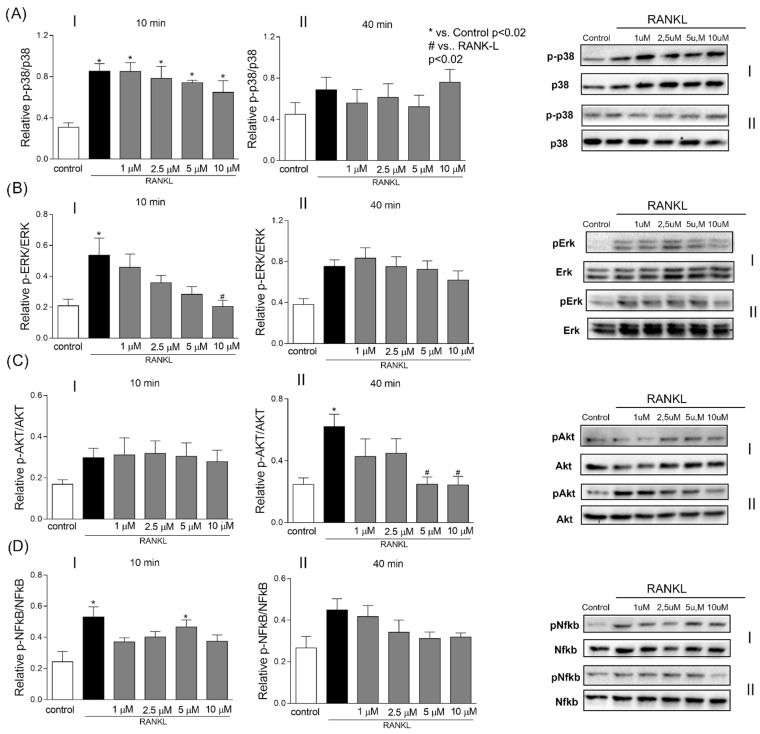
Chalcone T4 inhibits ERK and AKT activation. The graphs depict the mean and standard deviations of the densitometric analysis for the bands corresponding to ERK (**A**), AKT (**B**), p-38 (**C**) and NF-KB (**D**), after 10 (**I**) or 40 min (**II**) of stimulation with RANKL. Chalcone T4 suppressed ERK and AKT activation after 10 and 40 min, respectively, of stimulation with RANKL. Bars indicate mean results and vertical lines the standard error (SEM) of three duplicate experiments. *p* values are presented in the figure.

**Figure 5 ijms-24-07624-f005:**
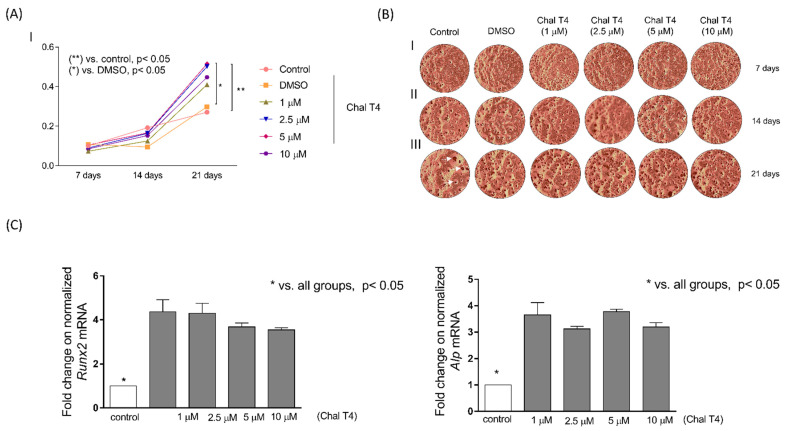
Chalcone T4 increases the formation of mineralized matrix nodules and gene expression of osteoblast markers. MC3TE-E1 were cultivated in osteogenic medium, and after 24 h treated with different concentrations of Chalcone T4. In (**A**), mean absorbance values at the end of each experimental period (7, 14 and 21 days). (**B**) Representative image of alizarin red staining. Images were registered on a digital inverted microscope (brightfield) (at a 40× magnification). The arrows highlight the formed calcification nodules. (**C**) Gene expression of *Alp* and *Runx2* in cells treated with the compound for 7 days. Bars indicate mean results and vertical lines the standard error (SEM) of three duplicate experiments.

## Data Availability

The data presented in this study are available on request from the corresponding author.

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
