# Peer review of "Chalcone T4 Inhibits RANKL-Induced Osteoclastogenesis and Stimulates Osteogenesis In Vitro"

_ijms, 2023, doi:10.3390/ijms24087624_

Round 1

Reviewer 1 Report

Review of “ Chalcone T4 Inhibits RANKL-induced Osteoclastogenesis and Stimulates Osteogenesis in vitro” Iolanda de Matos et al. 2023

Authors present interesting study regarding the impact of Chalcone T4 on osteoclast and osteoblast differentiation however the manuscript needs a lot of work regarding methods description and results presentation. The recommendations follow below:

Introduction:

The full name of genes or proteins should be mentioned first along with abbreviations in the text. In introduction some were added, other did not….

Methods:

Chalcone T4 synthesis:

The methods for synthesis of Chalcone T4 should provide more details or a bibliographic citation for similar synthesis methods. In addition, NMR and HPLC-DAD instruments experimental setups and brand references should be added.

Effect of Chalcone T4 on osteoclast differentiation:

In this experimental description are missing the references for products used in cell culture and experiments like alpha-MEM, FBS, DMSO, Pen/Strep, DAPI. This should be added.

In the experimental description is missing what compound was used  for cells fixation, please add.

Again, in several others sections the description of the references for several reagents and products are missing, please add.

Evaluation of cytokine gene expression (RT-qPCR):

A reference for comparative Δ(ΔCT) method is missing.

The sequence for the primers used to detect the expression of the gene of interest is missing. Please add primers sequences, even if only in supplementary data…

The full name of genes should be reference for the first time they are added before abbreviations and abbreviations should be in italic when reported for genes like Nfatc1, Oscar, Trap, Mmp-9 and Ctsk. In addition, for Runx2 gene authors have mention abbreviation with capital letters, this should only happen regarding proteins or human genes.

Statistical analysis:

Did the authors analyze the distribution of data obtained? This is missing in statistical descriptions, in order to apply One  Way ANOVA authors should make sure of that .

Results:

Results presented in figure 1:

In fluorescence microscope images presented is missing the scale bar for the photos. Please add.

The identification of the different times of treatment should be more evident, is not clear when figures are observed that results belong to different time treatments.

Results presented in figure 2:

Again the microscope images presented did not show the scale bar. In addition is not clear what should be the area of resorption. This is not evidence in photos. This should be revised.

Results presented in figure 5:

The asterisks presented in figure 5-B are not clear, in legend presented seems to indicate that is statistically different from control, however each graphic only present one asterisk over the control….

Discussion:

Authors should discuss how molecular mechanisms of Chalcones correlate with obtained results and pathways affected.

Author Response

Ref: Revised version of manuscript IJMS-2305709

Title: Chalcone T4 Inhibits RANKL-induced Osteoclastogenesis and Stimulates Osteogenesis in vitro. – Response to reviewer’s comments

Dear Reviewer,

We appreciate the time and consideration dedicated to read our manuscript and kindly invite you to consider our point-by-point response to your queries and comments below.

All changes in the revised manuscript are highlighted in yellow.

Reviewer #1:

  1. Comment: - Introduction: The full name of genes or proteins should be mentioned first along with abbreviations in the text. In introduction some were added, other did not….

Our reply/ answer: As recommended, the full name of genes and proteins was described first along with abbreviations.

  1. Comment: Methods: Chalcone T4 synthesis:

The methods for synthesis of Chalcone T4 should provide more details or a bibliographic citation for similar synthesis methods. In addition, NMR and HPLC-DAD instruments experimental setups and brand references should be added.

Our reply/ answer: As recommended, we added a reference on the synthesis process and experimental setups used.

  1. Comment: Effect of Chalcone T4 on osteoclast differentiation: In this experimental description are missing the references for products used in cell culture and experiments like alpha-MEM, FBS, DMSO, Pen/Strep, DAPI. This should be added. In the experimental description is missing what compound was used for cells fixation, please add. Again, in several others sections the description of the references for several reagents and products are missing, please add.

Our reply/ answer: In view of the reviewer's suggestion, all references to the reagents used in the experiments have carefully checked and added to the text when missing.

  1. Comment: Evaluation of cytokine gene expression (RT-qPCR): A reference for comparative Δ(ΔCT) method is missing.

The sequence for the primers used to detect the expression of the gene of interest is missing. Please add primers sequences, even if only in supplementary data…

The full name of genes should be reference for the first time they are added before abbreviations and abbreviations should be in italic when reported for genes like Nfatc1, Oscar, Trap, Mmp-9 and Ctsk. In addition, for Runx2 gene authors have mention abbreviation with capital letters, this should only happen regarding proteins or human genes.

Our reply/ answer: A table (supplementary table 1) with the sequence for the primers used to detect the genic expression was added, as suggested.

The full name of genes was described, and the abbreviations have been edited.

The reference for the Δ(ΔCT) method used in the analysis of RT-qPCR data is Livak & Schmittgen, Methods; v.25: 402-8, 2001 (PMID 11846609). This citation was added to the text as recommended.

  1. Comment: Statistical analysis: Did the authors analyze the distribution of data obtained? This is missing in statistical descriptions, in order to apply One Way ANOVA authors should make sure of that.

Our reply/ answer: Yes, the statistical analysis was performed according to the nature of the data distribution and considering the main scientific question posed in each experiment and the number of groups and periods compared. Information on data distribution analysis was added to the manuscript.

  1. Comment: Results: Results presented in figure 1:

In fluorescence microscope images presented is missing the scale bar for the photos. Please add.

The identification of the different times of treatment should be more evident, is not clear when figures are observed that results belong to different time treatments.

Our reply/ answer: A scale bar has been added to all images in figure 1b. In an attempt to clarify the difference between the periods of Chalcone treatment during osteoclastogenesis, we included the experimental periods in each bar graph and on the fluorescence image.

  1. Comment: Results presented in figure 2: Again the microscope images presented did not show the scale bar. In addition is not clear what should be the area of resorption. This is not evidence in photos. This should be revised.

Our reply/ answer: The scale bar was added to the images in figure 2, item b, and the resorption areas (i.e., the area in which the calcium phosphate coating was resorbed), was highlighted in one of the images to facilitate visualization and interpretation by reader. Pit assays are common experiments widely used in the assessment of osteoclast activity. Resorption of the calcium phosphate coating is visualized in a lighter shade of gray, since osteoclast-mediated removal of coating allows for increased translucency of the culture well in brightfield microscopy.   

  1. Comment: Results presented in figure 5: The asterisks presented in figure 5-B are not clear, in legend presented seems to indicate that is statistically different from control, however each graphic only present one asterisk over the control….

Our reply/ answer: We apologize for the mistake. The asterisk on the bar of the control group (without addition of Chalcone) indicates a statistically significant difference in relation to all other experimental conditions. We have corrected this mistake in the revised version.

  1. Comment: Discussion: Authors should discuss how molecular mechanisms of Chalcones correlate with obtained results and pathways affected.

Our reply/ answer: The biological mechanisms of chalcones, demonstrated in other in vitro studies, particularly their ability to modulate a series of targets of the inflammatory process involved in osteoclastogenesis and osteogenesis, were correlated with the findings in our study. The information added to the manuscript, as well as the relevant references, are highlighted in the discussion section.

Reviewer 2 Report

The study examines the effects of a newly synthesized chalcone, Chalcone T4, on osteoclastogenesis and osteogenesis. The authors synthesized the compound and tested its impact on pre-osteoclastic cells and pre-osteoblastic cells. The results indicate that Chalcone T4 decreased RANKL-induced osteoclast differentiation and activity, inhibited the expression of osteoclast-specific markers, and decreased ERK and AKT activation in a dose-dependent manner. Moreover, the compound stimulated osteogenesis by promoting the formation of mineralized matrix and the expression of ALP and RUNX2 in pre-osteoblastic cells. The authors also explore the possible mechanisms underlying the effects of Chalcone T4 on osteoclastogenesis and osteogenesis. These findings suggest that Chalcone T4 may be a promising candidate for the treatment of osteolytic diseases.

However, the manuscript is poorly written and organized, and the quality of the figures is low. Some significant limitations of the study include:

In Figure 1b, the fluorescence image does not correspond to the bar graph in Figure 1a. According to Figure 1b, 30 minutes before stimulation with RANKL, Chalcone T4 (1um) has fewer OC than Chalcone T4 (2.5um). However, in Figure 1a, 1um has more OC than 2.5um.

 In Figure 2a, the resorption area of the RANKL-only group is about 600μM in I, while the area of 2.5um in III is about 450. However, the representative image in Figure 2b shows that RANKL has clearly less resorption than 2.5um.

 The study should provide TRAP staining for verification of osteoclastogenesis.

 In Figure 4a WB image, the title is MANKL, and there are two p38 in time point II. Furthermore, in Figure 4c, the bands are outside of the rectangle.

The study should provide AR staining images.

Author Response

Ref: Revised version of manuscript IJMS-2305709

Title: Chalcone T4 Inhibits RANKL-induced Osteoclastogenesis and Stimulates Osteogenesis in vitro. – Response to reviewer’s comments

Dear Reviewer,

We appreciate the time and consideration dedicated to read our manuscript and kindly invite you to consider our point-by-point response to your queries and comments below.

All changes in the revised manuscript are highlighted in yellow.

Reviewer #2:

  1. Comment: In Figure 1b, the fluorescence image does not correspond to the bar graph in Figure 1a. According to Figure 1b, 30 minutes before stimulation with RANKL, Chalcone T4 (1um) has fewer OC than Chalcone T4 (2.5um). However, in Figure 1a, 1um has more OC than 2.5um.

Our reply/ answer: We agree with the reviewer and apologize for the mistake. A more representative image of the results found was added in figure 1b (I).

  1. Comment: In Figure 2a, the resorption area of the RANKL-only group is about 600μM in I, while the area of 2.5um in III is about 450. However, the representative image in Figure 2b shows that RANKL has clearly less resorption than 2.5um.

Our reply/ answer: A more representative image of the results was added in figure 2b.

  1. Comment: The study should provide TRAP staining for verification of osteoclastogenesis.

Our reply/ answer: TRAP is an enzyme abundantly produced by osteoclasts and therefore, widely used as a phenotypic marker of this cell type, however, TRAP staining is not the only reliable method for identifying osteoclasts. Moreover, it is possible to have TRAP-positive cells of small size, which are not compatible with the size of osteoclasts; thus in addition to TRAP expression/activity it is important to consider cell size and multinucleation as criteria for identifying osteoclasts in vitro. The experimental approach used allows for the verification of a cytoskeleton change associated with osteoclast differentiation, for the verification of cell size and for the presence of three or more nuclei in each cell. In support to our approach, biologically it is known that after differentiation, osteoclasts strongly attach to the bone matrix and form a bone resorption area, where metalloproteases and matrix acids are secreted by osteoclasts for bone resorption (PMID: 10428500). To promote the sealing of the bone matrix, during the terminal differentiation phase, osteoclasts form a structure similar to a ring in their periphery, containing numerous actin filaments, with an important role in the activity of osteoclastic bone resorption, and which serves as a maturation marker of formed osteoclasts (PMID: 28576497). Cell staining with phalloidin conjugated to a fluorophore, and association with a genomic DNA marker, allows the identification and quantification of osteoclasts by fluorescence, through the visualization of an actin ring containing three or more nuclei, being, therefore, a reliable and extensively used to identify osteoclasts. In addition to identifying the presence of osteoclasts, our study also evaluated the effect of the compound on osteoclastic activity by using a pit assay, since the number of osteoclasts may be similar, but the resorptive activity may be significantly different, or vice versa. In addition, since the literature has shown that the increase in osteoclastic differentiation and activity is related to the increase in the expression of certain biological markers (Cathepsin-K, Mmp-9, Nfatc1, Oscar, TRAF6, c-fos) and activation of intracellular signaling pathways (PMID: 12748652), we also evaluated the effect of Chalcone T4 on the expression of osteoclastic markers and on the activation of intracellular signaling pathways (MAPKs, AKT, NF-kB). We think that collectively the results are sufficient to indicate the potential of the compound in reducing osteoclastogenesis and resorptive activity.

  1. Comment: In Figure 4a WB image, the title is MANKL, and there are two p38 in time point II. Furthermore, in Figure 4c, the bands are outside of the rectangle.

Our reply/ answer: We appreciate the attentive review. The mistakes in figure 4 have been corrected.

  1. Comment: The study should provide AR staining images.

Our reply/ answer: Representative image of alizarin red staining was added to Figure 5 of the manuscript, as suggested by the reviewer.

Round 2

Reviewer 1 Report

Review of “Chalcone T4 Inhibits RANKL-induced Osteoclastogenesis and Stimulates Osteogenesis in vitro” by Iolanda Matos et al 2023

Authors have corrected and included all the changes recommended by the reviewers, only small details regarding gene nomenclature need to be corrected.

In figure 3 description authors mention the detection of TRAP transcripts, which is in capital letters, authors forget to change for mouse nomenclature, in addition, the official name for the gene that can promote TRAP activity is Acp5 (Acid Phosphatase 5, Tartrate Resistant), recommend to use this abbreviation.

In discussion authors mention that “It is important to consider the limitations of the approach experimental, since we only evaluated the transcriptional levels (mRNA) of NFATc1, it is possible that Chalcone T4 has modulated the protein levels of this transcription factor”, author should adjust abbreviation to murine notation, Nfatc1….

In discussion authors mention “ Reduction of mRNA levels and protein of Cathepsin k, cfos, TRAF6 and TRAP were observed in macrophages treated with natural chalcones” in this case, abbreviations for the genes should be added in italic….

Author Response

Ref: Revised version of manuscript IJMS-2305709

Title: Chalcone T4 Inhibits RANKL-induced Osteoclastogenesis and Stimulates Osteogenesis in vitro. – Response to reviewer’s comments

Dear Reviewer,

We greatly appreciate all comments, which have thoroughly addressed in the revised version of the manuscript.

Changes recommended by reviewer #1 are highlighted in yellow.

We also revised the sections (abstract, introduction, results, and discussion) in order to make the text clearer, and properly written. All changes are tracked so that reviewers can identify them.

Reviewer #1:

  1. Comment: - Introduction: In figure 3 description authors mention the detection of TRAP transcripts, which is in capital letters, authors forget to change for mouse nomenclature, in addition, the official name for the gene that can promote TRAP activity is Acp5 (Acid Phosphatase 5, Tartrate Resistant), recommend to use this abbreviation.

Our reply/ answer: As recommended, we changed the name of the gene to Acp5, and followed the mice nomenclature (capital letter and in italics).

  1. Comment: In discussion authors mention that “It is important to consider the limitations of the approach experimental, since we only evaluated the transcriptional levels (mRNA) of NFATc1, it is possible that Chalcone T4 has modulated the protein levels of this transcription factor”, author should adjust abbreviation to murine notation, Nfatc1….

Our reply/ answer: Thanks for the careful review. The gene name has now been appropriately changed.

  1. Comment: In discussion authors mention “ Reduction of mRNA levels and protein of Cathepsin k, cfos, TRAF6 and TRAP were observed in macrophages treated with natural chalcones” in this case, abbreviations for the genes should be added in italic….

Our reply/ answer: Abbreviations for genes are added in italics.

Reviewer 2 Report

There are no questions left. This is now a better revised manuscript.

Author Response

Ref: Revised version of manuscript IJMS-2305709

Title: Chalcone T4 Inhibits RANKL-induced Osteoclastogenesis and Stimulates Osteogenesis in vitro. – Response to reviewer’s comments

Dear Reviewer,

We greatly appreciate all comments, which have thoroughly addressed in the revised version of the manuscript.

We revised the sections (abstract, introduction, results, and discussion) in order to make the text clearer, and properly written. All changes are tracked so that reviewers can identify them.